# Multi-Drug Resistance to *Salmonella* spp. When Isolated from Raw Meat Products

**DOI:** 10.3390/antibiotics11070876

**Published:** 2022-06-29

**Authors:** Joanna Pławińska-Czarnak, Karolina Wódz, Magdalena Kizerwetter-Świda, Janusz Bogdan, Piotr Kwieciński, Tomasz Nowak, Zuzanna Strzałkowska, Krzysztof Anusz

**Affiliations:** 1Department of Food Hygiene and Public Health Protection, Institute of Veterinary Medicine, Warsaw University of Life Sciences, Nowoursynowska 159, 02-776 Warsaw, Poland; janusz_bogdan@sggw.edu.pl (J.B.); z.strzalkowska@gmail.com (Z.S.); krzysztof_anusz@sggw.edu.pl (K.A.); 2Laboratory of Molecular Biology, Vet-Lab Brudzew, Ul. Turkowska 58c, 62-720 Brudzew, Poland; karolina.wodz@labbrudzew.pl (K.W.); vetlab@interia.pl (P.K.); tomasz@labbrudzew.pl (T.N.); 3Department of Preclinical Sciences, Institute of Veterinary Medicine, Warsaw University of Life Sciences-SGGW, Ciszewskiego Str. 8, 02-786 Warsaw, Poland; magdalena_kizerwetter_swida@sggw.edu.pl

**Keywords:** *Salmonella enterica*, Enteritidis, multidrug-resistant, Derby, foodborne pathogens

## Abstract

*Salmonella* spp. is the most frequent cause of foodborne diseases, and the increasing occurrence of MDR strains is an additional and increasing problem. We collected *Salmonella* spp. strains isolated from meat (poultry and pork) and analysed their antibiotic susceptibility profiles and the occurrence of resistance genes. To determine the susceptibility profiles and identify MDR strains, we used two MIC methods (MICRONAUT and VITEC2 Compact) and 25 antibiotics. Phenotypic tests showed that 53.84% strains were MDR. Finally, molecular analysis strains revealed the presence of *bla*_SHV_, *bla*_PSE-1_, *bla*_TEM_, but not *bla*_CTX-M_ genes. Moreover, several genes were associated with resistance to aminoglycosides, cephalosporins, fluorochinolones, sulfonamides, and tetracyclines. This suggests that further research on the prevalence of antibiotic resistance genes (ARGs) in foodborne strains is needed, especially from a One Health perspective.

## 1. Introduction

The annual report on trends and sources of zoonoses published in December 2021 by the European Food Safety Authority (EFSA) and the European Centre for Disease Prevention and Control (ECDC) shows that nearly one in four foodborne outbreaks in the European Union (EU) in 2020 were caused by *Salmonella* spp., which makes this bacteria the most frequently reported causative agent for foodborne outbreaks (694 foodborne outbreaks in 2020) [1].

In the EU 52,702 confirmed cases of salmonellosis in humans were reported and salmonellosis remains the second most commonly reported zoonosis in humans after campylobacteriosis. The three most commonly reported *Salmonella enterica* subsp. *Enterica* serovars in 2020 were *S.*
*Enteritidis*, *S.*
*typhimurium*, and monophasic *S.*
*typhimurium*, representing 72.2% of confirmed human cases with known serovar in 2020. Most of the reported salmonellosis foodborne outbreaks were caused by *S.*
*Enteritidis* serovar (57.9%). *S.*
*Enteritidis* was the predominant serovar in both human salmonellosis cases and reported foodborne outbreaks. Due to the COVID-19 pandemic, total numbers of reported salmonellosis cases as well as foodborne outbreaks are lower compared to previous years’ data. Increased use of hygiene equipment, reduced exposure to food served in restaurants and canteens, and more frequent cleaning during domestic food preparations might have had an impact on reported data on salmonellosis. Despite the facts above, trends in salmonellosis occurrence since 2016 data did not reveal statistically significant changes (EFSA December 2021) [1].

Bacteria of the genus *Salmonella* are gram-negative, mostly motile rods, belonging to the *Enterobacteriaceae* family. *Salmonella* spp. is well-established as a pathogen causing gastrointestinal diseases in humans and animals all over the world. Two species are included in the genus *Salmonella*: *Salmonella enterica* spp. and *Salmonella bongori* spp. Almost 99% of the *Salmonella* strains that cause infections in humans or other warm-blooded animals belong to the species *S. enterica*, which includes six subspecies and >2587 serovars [2].

*Salmonella enterica* subsp. *Enterica* includes approximately 1547 serotypes which can cause infections in animals and humans [2]. *Salmonella* infections in humans are usually caused by eating food of animal origin, mostly eggs, poultry meat, or pork [3,4]. The analysis by Gutema et al. (2019) shows that beef and veal can also be a source of *Salmonella* spp. infection due to these animals being potential asymptomatic carriers [3]. 

Currently, one of the most important health problems in the world is the antimicrobial resistance of *Salmonella* spp. [4,5]. Data from the EU show that the occurrence of resistance in *Salmonella* from pigs, cattle, and broiler chickens largely resembles the appearance of resistance reported for *Salmonella* in various foodstuffs and in people (EFSA [4]).

Multi-drug resistant *Salmonella* constitutes a serious threat to public health through food-borne infections [6,7,8]. Currently, such multi-drug resistant strains are increasingly isolated from beef and pork [9,10] poultry [11].

Because the problem of antimicrobial resistance became a global problem, in 2003 WHO, together with the Food and Agriculture Organization of the United Nations (FAO) and the World Organization for Animal Health (OIE), began work on creating a List of Critically Important Antimicrobials for Human Medicine (WHO CIA List) [12]. Tacconelli et al. in 2018, pointed out that global research and development strategies should also include antibiotics active against more common community bacteria, such as *Salmonella* spp., *Campylobacter* spp. and *H. pylori*, which are resistant to antibiotics [13]. Therefore, the scope of the new edition of the WHO CIA List, published in 2019, is limited to antibacterial drugs of which most are also used in veterinary medicine. It is very important to use critically important antimicrobials the most prudently in human and veterinary medicine. With accordance monitoring of antimicrobial resistance in food and food-producing bacteria, as defined in Commission Implementing Decision 2013/652/EU, *Salmonella* antibiotics resistance, isolated from food and food-producing animals, should be targeted at broilers, fattening pigs, calves less than 1 year old, and their meat (CID 2013/652/EU).

The aim of our research is to determine the antibiotic resistance of *Salmonella* spp. isolated from raw meat products from beef, pork, and poultry production plants.

## 2. Results

Of the 170 meat samples tested, no *Salmonella* spp. were found in beef samples; but, three *Citrobacter braakii* were isolated from them. Only one of the pork samples was positive for *Salmonella* spp. and three *Citrobacter braakii* were isolated from them. Details of any identification difficulties during the isolation of *Salmonella* spp. from meat samples tested were presented by Pławińska-Czarnak in 2021 [14]. From the poultry samples, 38 were positive for *Salmonella* spp. All *Salmonella* strains of the isolated species belong to *Salmonella enterica* subsp. *enterica* and represented seven serotypes which shown in Table 1. 

The most common serovars from all positive samples were: *S*. Enteritidis (58.97%); *S*. Derby (12.82%) and *S*. Newport (12.82%), which were less frequently isolated; *S*. Infantis (5.13%); *S*. Kentucky (5.13%); *S*. *indiana* (2.56%); and *S*. Mbandaka (2.56%) (the details of the results are presented in Table 1).

### 2.1. Antibiotic Susceptibility

Antibiotic susceptibility testing conducted on the 39 *Salmonella* strains shows that only one strain (*S.*
*Enteritidis*) has resistance to two classes of antibiotics (CPH-GEN-STR) whereas 38 strains (64%) were resistant to one or more of the tested antibiotics. However, no resistance against imipenem or colistin was detected. Surprisingly, we detected that 100% of *Salmonella* strains were phenotypically resistant to streptomycin and gentamycin. *Salmonella* strains had intermediate resistance to: amoxicillin (5.13%, *S.*
*kentucky*, *S.* Newport), cephalexin (30.77%, *S.* Infantis, *S.*
*Enteritidis*), ceftiofur (2.56%, *S*. Infantis), neomycin (7.96%, *S.* Newport), enrofloxacin (23.08%, *S.* Infantis, *S.* Mbandaka, *S.* Newport, *S.*
*Enteritidis*), norfloxacin (15.8%, *S.*
*derby*, *S*. *indiana*, *S*. Enteritidis), doxycycline and oxytetracycline (5.13%, *S*. Derby, *S*. Enteritidis), florfenicol (56.41%, *S*. Mbandaka, *S*. Kentucky, *S*. Newport, *S*. Enteritidis), and trimethoprim-sulfamethoxazole (2.26%, *S*. Derby). In total, 35.9% (14/39) of the strains were resistant to ampicillin, 38.46% (15/39) to amoxicillin, and 7.69% (3/39) to amoxicillin and clavulanic acid. In the case of cephalosporins 46.15% (18/39) of the strains were resistant to cephalexin, 38.46% (14/39) to cefalotin, 97.43% (38/39) to cefapirin, 17.95% (7/39) to cefoperazone, 23.08% (9/39) to ceftiofur, and 12.82% (5/39) to cefquinome. In the case of aminoglycosides, 10.25% (4/39) were resistant to neomycin. In the case of fluoroquinolones, 28.2% (11/39) were resistant to enrofloxacin, 82.05% (32/39) to flumequine, 33.33% (13/39) to marbofloxacin, and 10.25% (4/39) to norfloxacin. A total of 25.64% (10/39) were resistant to tetracyclines, 38.46% (14/39) to florfenicol, 56.41% (22/39) to lincomycin/spectinomycin, and 7.69% (3/39) to trimethoprim/sulfamethoxazole.

### 2.2. Prevalence of Multiple Drug Resistance

In our study, most of *S*. Enteritidis showed an MAR index lower than 0.3, whereas one (*S*. Newport) showed an MAR index above 0.5. We observed a high prevalence of multiple antibiotic resistance amongst the isolates where 53.84% of the isolates were MDR strains, with resistance from three to six different classes of antibiotics.

### 2.3. Antimicrobial Resistance Profile

All *Salmonella* strains of the isolated species belongs to *Salmonella enterica*
*subsp*. *enterica* and represented seven serotypes (Derby, *indiana*, Infantis, Mbandaka, Kentucky, Newport, and Enteritidis). All isolated *Salmonella* were sensitive to imipenem (IMP) and colistin (COL)/polymixin B (PB).

A total of 53.84% *Salmonella* spp. strains isolated from meat were classified as MDR strains that were resistant to the six antibiotic classes: penicillins, cephalosporins, aminoglycosides, fluorochinolones, sulfonamides, and tetracyclines. *S*. Newport (sample 1) presented the most extensive resistance profiles to 17 antibiotics (AMP-AMX-AMX/CL-CFX-CFT-CPH-GEN-NEO-STR-ENR-UB-MRB-NOR-DOX-OXY-TET-LIN/SP), belonging to 5 classes of antibiotics (β-lactams, aminoglycoside, fluorochinolones, tetracyclines and lincosamides with spectinomycin. In one of *S*. Derby (AMP-AMX-CFX-CFT-CPH-CFP-CFTI-CFQ-GEN-STR-ENR-UB-MRB-FLR-LIN/SP-TR/SMX) and *S*. Newport (AMP-AMX-AMX/CL-CFX-CFT-CPH-GEN-STR-ENR-UB-MRB-NOR-DOX-OXY-TET-LIN/SP), extensive resistance profiles to 16 antibiotics were present. In *S.*
*indiana* (AMX-AMX/CL-CTX-CPH-CFTI-GEN-NEO-STR-DOX-OXY-TET-FLR-LIN/SP-TR/SMX), extensive resistance profiles to 14 antibiotics were present.

The classes to which it presented the highest resistance were β-lactams (AMP, AMX) and beta-lactam/beta-lactamase inhibitor combination (AMX/CL), I generation cephalosporin (CFX-CFT-CPH), III generation cephalosporin (CFTI, CFP), aminoglycosides (GEN-NEO-STR), fluorochinolones (ENR-UB-MRB-NOR), and tetracyclines (DOX-OXY-TET). The most diverse serotype in terms of antimicrobial resistance turned out to be *S*. Enteritidis, in which 13 patterns of resistance were observed. Serovar *S.* Mbandaka showed complete resistance to 9 antibiotics (AMP-AMX-CFX-CFT-CPH-GEN-STR-UB-LIN/SP), and *S.* Infantis showed resistance to 10 antibiotics to varying degrees. The least resistant strain of *S*. Enteritidis was strain from pork meat resistant to 3 antibacterial substances (CPH-GEN-STR), and the most resistance to *S*. Enteritidis was strain 11 from poultry meat (AMP-CFX-CFT-CPH-CFTI-GEN-STR-UB-MRB-FLR-LIN/SP).

For the particular serotypes of *Salmonella enterica* spp. *enterica*, all individual patterns of resistance to multiple antibiotics are presented in Table 2.

The isolates were subjected to antibiotic susceptibility tests against 33 antibiotics belonging to ten different classes using the MIC method Merlin MICRONAUT (MERLIN Diagnostika GmbH, Niemcy) and AST-GN96 CARD and VITEK2 system (Biomerieux, Marcy-l’Étoile, France). The AST card is essentially a miniaturised and abbreviated version of the doubling dilution technique for MICs determined by the microdilution [15]. The multiple antibiotics resistance index (MAR) was performed for isolates showing resistance to more than two antibiotics and is presented in the Table 2 [16]. 

### 2.4. Genotypic Resistance

The gene *bla*_CMY-2_ that confers resistance to cefoperazone/ceftiofur was detected in 41.02%, and *bla*_SHV_ in 35.9%. of strains. However, some *Salmonella* spp. strains did not exhibit phenotypic resistance to III generation cephalosporins. In addition, 30.77% of the strains demonstrated the presence of the genes *bla*_PSE-1_ and 48.72% *bla*_TEM_ that conferred resistance to ampicillin. Most of ampicillin-resistant strains (85.71%) contained *bla*_PSE-1_ and *bla*_TEM_, and 14.28% harboured only *bla*_TEM_ gene. The gene *aadB* was detected in eight strains, mainly in *S*. Derby. However, all *Salmonella* spp. strains were phenotypically resistant to gentamicin. The genes *aadA*, *strA*/*strB* that confers resistance to streptomycin was detected in all strains. All of neomycin resistant strains carried *aphA1* and *aphA2* genes. The *tetA* and *tetB* genes were detected in all strains resistant to doxycycline and oxytetracycline. Sulphonamide-resistant strains contained at least one *sul* (1, 2, 3) and *adfR* gene, of which the *sul2* and *adfR1* were the most frequently detected genes. The gene *floR,* that confers resistance to florfenicol, was detected in all strains resistant to florfenicol.

Distribution of the various resistance genes and the prevalence of the corresponding serovars are shown in Table 3.

## 3. Materials and Methods

### 3.1. Sampling

A total number of 190 raw meat samples (60 beef, 60 pork, and 70 poultry) were obtained from three sources within the meat industry, such as cuttings of beef, pork and poultry carcasses in central Poland. All samples were obtained from carcass parts of animals recognised as healthy: the tissues and organs of which were classified by the veterinary inspection as fit for human consumption. All samples were considered a single sample, weighing at least 200 g for each type of meat. The meat samples were collected randomly, using an aseptic technique and packed into sterile bags, which were labeled. All samples were transported to the laboratory in refrigerated containers at a temperature 4 °C and processed within five hours.

### 3.2. Salmonella spp. Isolation and Identification 

*Salmonella* spp. from all samples were isolated in accordance with PN-EN ISO 6579-1:2017-04 Microbiology of the food chain—Horizontal method for the detection, enumeration and serotyping of *Salmonella*—Part 1: Detection of *Salmonella* spp. (ISO 6579-1:2017). Samples were pre-enriched: for pork and beef samples, the 10 g of each sample was mixed with 90 mL Buffered Pepton Water (GRASO, Gdansk, Poland), and the 25 g of each poultry meat sample was mixed with 225 mL BPW with a temperature of 25 °C (±3 °C) in a sterile stomacher bag (Whirl-Pak, Nasco, Madison, WI, USA), and crushed for 2 min. After that, they were incubated at 37 °C for 18 h. Selective proliferation of *Salmonella* spp. was carried out using the MSRV agar (Modified semi-solid Rappaport-Vassiliadis—MSRV agar, GRASO, Poland) with 0.1 mL of the pre-enriched culture as three equally spaced spots on the surface of the MSRV agar were incubated at 41.5 °C for 24 h and 1 mL of the culture obtained was put to a tube containing 10 mL of Muller-Kauffmann tetrathionate-novobiocin (MKTTn) broth (GRASO, Gdansk, Poland) and incubated at 37 °C for 24 h. From the positive growth obtained on the MSRV agar, it was chosen as the furthest point of opaque growth from the inoculation points, and picked up a 1 μL loop and was inoculated on two selective agars: XLD (Xylose Lysine Deoxycholate agar, GRASO, Gdansk, Poland) and BGA (Brilliant Green agar, OXOID, Hampshire, UK). From the liquid culture obtained in the MKTTn, broth was picked up of a 10 μL loop and spread on XLD agar and BGA agar to obtain well-isolated colonies. All selective agars were incubated at 37 °C for 24 h (±3 h). *Salmonella*-suspect colonies were transferred to Nutrient agar (GRASO, Gdansk, Poland) to obtain the pure culture for further testing.

#### 3.2.1. DNA Preparation and Presumptive Salmonella Confirmation

The Real-time PCR method, and an amplification based on detection gene specific for *Salmonella*, was used to confirm presumptive identification. DNA for real-time PCR was extracted from bacterial cells, using commercial Kylt^®^ DNA Extraction-Mix II (Anicon, Emstek, Germany). For the detection of *Salmonella* spp. *commercial* Kylt^®^ *Salmonella* spp. (Anicon, Germany) was used, and for the simultaneous detection of *Salmonella* Enteritidis, the Typhimurium commercial Spp-Se-St PCR (BioChek, Reeuwijk, The Netherland) kit was used. The Real Time PCR method to detect *Salmonella* was performed according to the manufacturer’s instructions with using Applied Biosystems 7500 Fast Real-Time PCR System (Thermo, Waltham, MA, USA).

#### 3.2.2. Biochemical Strain Identification

For identification of the strains, two commercially available biochemical tests were used according to the manufacturer’s instructions: Api20E (BioMérieux, Marcy-l’Étoile, France) and the VITEK^®^ 2 GN cards (Biomerieux, Marcy-l’Étoile, France). 

#### 3.2.3. Serological Testing

Serotyping was performed according to the White-Kauffmann-Le Minor scheme. Serological testing was carried out by slide agglutination with commercial H poly antisera to verify the genus of *Salmonella enterica* (IBSS Biomed, Lublin, Poland), O group antisera to determine the O group, (IBSS Biomed, Poland), and H phase and H factor antisera to determine the H phase and H factor (IBSS Biomed, Lublin, Poland, Bio-Rad, Chercules, CA, USA), as described in Pławińska-Czarnak [17].

### 3.3. Antimicrobial Sensitivity Testing

Each *Salmonella* strain was first subcultured as described previously. From an 18–24 h culture, a DensiCHEK Plus (Biomerieux, Marcy-l’Étoile, France) instrument was used to perform a suspension with a 0.5 McFarland range. Then, 145 μL of this inoculum was transferred to another VITEK^®^ tube containing 3 mL 0.45% saline. The card was automatically filled by a vacuum device and automatically sealed. It was manually inserted in the VITEK2 Compact reader-incubator module, and every card was automatically subjected to a kinetic fluorescence measurement every 15 min. This is an automated test methodology based on the MIC technique reported by MacLowry and Marsh [18], and Gerlach [19]. A loop of the suspension was also inoculated onto blood agar (GRASO, Poland) for the purity check.

Antimicrobial susceptibility was assessed by determining the MIC values using a 96 well MICRONAUT Special Plates with antimicrobials: β-lactams/aminopenicillin (amoxicillin—AMX, amoxicillin and clavulanic acid—AMX/CL), β-lactams/I generation cephalosporins (cephalexin—CFX, cephapirin—CPH), β-lactams/III generation cephalosporins (ceftiofur—CFTI), β-lactams/IV generation cephalosporins (cefquinome—CFQ), β-lactams/penicillin cloxacillin—CLO, penicillin G—PG, nafcillin—NAF), aminoglycoside (gentamicin—GEN, neomycin—NEO, streptomycin—STR), polymyxins (colistin—COL), fluorochinolones (enrofloxacin—ENR, norfloxacin—NOR), tetracyclines (doxycycline—DOX, oxytetracycline—OXY), macrolides erythromycin—ERY, tylosin—TYL), florfenicol—FLR), lincosamides (lincomycin—LIN, lincomycin/spectinomycin—LIN/SP), trimethoprim-sulfamethoxazole—TR/SMX, tiamulin—TIA, tylvalosin—TYLV (MERLIN Diagnostika GmbH, Bremen, Niemcy). Simultaneously, antimicrobial susceptibility was assessed by determining the MIC values using a VITEK^®^ 2 System and AST-GN96 cards for Gram-negative bacteria (BioMérieux). The AST card is essentially a miniaturised and abbreviated version of the doubling dilution technique for MICs determined by the microdilution method [c]. 

The MERLIN antibiotics concentration (µg/mL) is as follows: amoxicillin—0.25, 2, 4, 8, 16; amoxicillin and clavulanic acid—4/2, 8/4, 16/8; cephalexin—8, 16; cephapirin—8, ceftiofur—2; cefquinome—2, 4; cloxacillin—2; penicillin 0.0625, 0.125, 2, 8; nafcillin—2; gentamicin—4, 8; neomycin—8; streptomycin—8; colistin—2; enrofloxacin—0.5, 2; norfloxacin—1, 2; doxycycline—2, 4, 8; oxytetracycline—2, 4, 8; erythromycin—0.25; 0.5, tylosin—TYL; florfenicol—2, 4; lincomycin—2, 8; lincomycin/specinicin—8, 32; trimethoprim-sulfamethoxazole—2/38; tiamulin—16; and tylvalosin—2, 4.

With using AST-GN96 susceptibility for β-lactams/aminopenicillin (ampicillin—AMP, amoxicillin and clavulanic acid—AMX/CL), β-lactams/I generation cephalosporins (cefalexin -CFX), β-lactams/III generation cephalosporins (cefalotin—CFT, cefoperazone CFP), β-lactams/III generation cephalosporins (ceftiofur—CFTI), β-lactams/IV generation cephalosporins (cefquinome—CFQ), carbapenems (imipenem—IPM), polymyxin (polymixin B -PB), aminoglycoside (gentamicin—GEN, neomycin—NEO), fluorochinolones (enrofloxacin—ENR), flumequine—UB), marbofloxacin—MRB), tetracycline -TET, florfenicol—FLR, and trimethoprim/sulfamethoxazole (TR/SMX), were assessed.

The AST-GN96 antibiotics concentration (µg/mL) is as follows: ampicillin—4, 8, 32; amoxicillin and clavulanic acid—4/2, 16/8, 32/16; cephalexin—8, 16, 32; efalotin—2, 8, 32; cefoperazone 4, 8, 32; cefquinome—0.5, 1.5, 4; imipenem 1, 2, 6, 12; polymixin B 0.25, 1, 4, 16; gentamicin—4, 16, 32; neomycin—8, 16, 64; enrofloxacin—0.25, 1, 4; flumequine—2, 4, 8; marbofloxacin—1, 2; tetracycline—2, 4, 8; florfenicol—1, 4, 8; trimethoprim/sulfamethoxazole—1/19, 4/76, 16/304.

The MICs were interpreted according to Clinical and Laboratory Standards Institute (CLSI) and FDA breakpoints (CLSI M100-ED28, 2018). The AST card is essentially a miniaturised and abbreviated version of the doubling dilution technique for MICs determined by the microdilution method. 

### 3.4. Determination of Antibiotics Resistance Profile of Salmonella spp. Isolates

In order to calculate multiple antibiotics resistance, we used the formula according to the Akinola 2019, MAR index [16]:$$\mathrm{MAR}=\frac{\mathrm{Number}\;\mathrm{of}\;\mathrm{resistance}\;\mathrm{to}\;\mathrm{antibiotics}}{\mathrm{Total}\;\mathrm{number}\;\mathrm{of}\;\mathrm{antibiotics}\;\mathrm{tested}}$$

#### Detection of Antimicrobial Resistance Genes by PCR

Mueller–Hinton agar was used to culture the bacterial isolates overnight at 35 °C. Bacterial DNA isolation was performed using a standard bacterial DNA isolation Kylt^®^ DNA Extraction-Mix II (Anicon, Emstek, Germany). Eighteen resistance genes (*aadA*, *strA/strB*, *aphA1*, *aphA2*, *aadB*, *tetA*, *tetB*, *sul1*, *sul2*, *sul3*, *dfrA1*, *dfrA10*, *dfrA12*, *floR*, *bla*_TEM_, *bla*_SHV_, *bla*_CMY-2_, *bla*_PSE-1_ and *bla*_CTX-M_) were analysed by conventional PCR, using specific primer pairs in multiplex or a single PCR reaction. The primer sequences predicted PCR product sizes and references shown in Table 4.

### 3.5. Statistical Assessment

Statistical testing was performed with Statistica software, version 13.1. Descriptive statistics were computed to determine the proportions of isolates resistant to different antimicrobial agents. Chi square tests were adopted for the determination of statistical significance of differences between the proportions.

## 4. Discussion

Our data show that poultry meat is a relevant source of *Salmonella*, and the prevalent serovar was Enteritidis (56.41%). We estimate the antibiotic susceptibility profiles of *Salmonella* strains, and we found a high rate of strains showing at least one phenotypic resistance. In our study, sensitivity to 25 antibiotics were assessed. Penicillins (cloxacillin, penicillin G, nafcillin), macrolides (erythromycin, tylvalosin), lincomycin, tiamulin, and tylvalosin were excluded from analysis, due to a natural lack of activity against *Salmonella*.

The results of the antibiotic resistance indicate that the *Salmonella* spp. strains isolated from meat can be categorized as resistant to MDR: that is, bacteria exhibiting resistance to one or more antibiotics from three or more classes of antibiotics. These bacteria are resistant to β-lactams, aminoglycosides, cephalosporins, fluorochinolones, sulfonamides, and tetracyclines. Resistance to third generation cephalosporins exhibited by the strains isolated from meats represents a concern, because these antibiotics are used for salmonellosis treatment in human, thus rendering the transmission of resistant bacteria a public health problem. All strains isolated from meat were resistant to gentamycin, which is one of the major antibiotics used in the treatment of urinary infections in humans, and were resistant to streptomycin used to treat tuberculosis and *Burkholderia* infection. Although streptomycin is an aminoglycoside and not used for *Salmonella* treatment, streptomycin resistance has been widely used as an epidemiological marker. Resistance to streptomycin is analogous to the phenotypic characteristics observed in multi-drug resistance to ampicillin, chloramphenicol, streptomycin, sulfonamides, and tetracyclines [23,24]. Regarding the resistance to ampicillin (35.89%), previous studies from different countries report highest resistance rates [25]. 

Moreover, *Salmonella* Derby from meat shows resistance to cefequinome, fourth generation cephalosporins, and antibiotics used in the treatment of mastitis and bovine pneumonia. In *Salmonella* Derby and *indiana* (both in the BO4 group), we found resistance against sulphonamides, a class of antibiotics used in severe *Salmonella* infections. We also observed resistance to third generation cephalosporins (cefoperazone and ceftiofur) in four *Salmonella* Derby strains isolated from poultry meat. In addition, a high percentage of strains (Indiana, Infantis, Kentucky, and Newport) showed resistance to tetracyclines (24.64%), despite the fact that, in 2006, the European Union, imposed a ban on the non-therapeutic use of antibiotics important to humans, such as tetracyclines, in animal treatment. A total of 53.84% of tested strains showed an MDR profile with resistance to one or more antibiotics from three or more classes of antibiotics. On the other hand, all the *Salmonella* spp. strains were susceptible to imipenem, which is similar to the result reported previously [26]. Carbapenems are the final choice of antibiotics used in the treatment of salmonellosis when the bacteria exhibit resistance to antibiotics, such as ciprofloxacin and third generation cephalosporins.

These data are alarming for consumers because of the real possibility of an infection with an MDR strain in food, but also because these strains showed resistance to antibiotic classes crucial in human medicine, such as beta-lactamases.

Finally, because these antibiotic phenotypes can be conferred by several ARGs, the detection of resistance genes was performed in order to confirm phenotypic pattern.

In *Salmonella*, the main mechanism of resistance to β-lactams is the acquisition *bla* gene encodes beta-lactamase hydrolytic enzymes, which inactivate the antibiotic [27]. Extended-spectrum beta-lactamases (ESBLs), which inactivates first-, second-, and third-generation cephalosporins and penicillins, and are encoded multi-variant *bla*_TEM_, *bla*_SHV_ and *bla*_CTX-M_ genes [28]. The *bla*_CTX-M_ genes encode for the extended-spectrum of β-lactamases (ESBLs) were not present in analysed strains. These types of β-lactamases are active against cephalosporins and monobactams (but not carbapenems), and are currently of great epidemiological and clinical interest. The *bla*_SHV_ gene was found to be the most prevalent gene amongst our isolates, mainly in *S.*
*Enteritidis*. The *bla*_SHV_ gene is associated with *Enterobacteriaceae* in causing nosocomial infections, but also in isolates from different sources (human, animal, and environment). The gene *bla*_CMY-2_ encodes an extended-spectrum beta-lactamase that is responsible for hydrolyzing the β-lactam ring that was detected in 35.89% of strains. However, some *Salmonella* spp. strains did not expose phenotypic resistance to this antibiotic. This gene confers resistance to ampicillin, ceftiofur, cefoperazone and is associated with mobile elements, thus increasing the probability of transmission between bacteria [29]. In our study, 28.21% of the strains demonstrate the presence of the genes blaPSE-1 and *bla*_TEM_ that encode β-lactamases that confer resistance to ampicilin. In a study conducted in Colombia, 69.4% of the strains isolated from broiler farms had both genes; thus, a frequency was higher than that found in the present study [30]. Five *S.*
*derby*, one *S.*
*Enteritidis*, and all *S.*
*kentucky* that were phenotypically resistant to ampicillin and third generation cephalosporins, showed the presence of the genes *bla*_PSE-1_, *bla*_TEM_, *bla*_CMY-2_, but not and *bla*_CTX-M_. The streptomycin resistance gene *aadA* and *strA/strB* were detected in all of the strains. Interestingly, White et al. [31] showed that *Salmonella* strains isolated from meat that had the *aadA* genes but were susceptible to streptomycin, probably due to gene silencing. The gene *sul2* encodes DHPS (dihydropteroate synthase) was found in 7.69% of the strains (*S.*
*derby* and *S.*
*indiana*). In a previous study, the gene *sul1* is reported to be the most prevalent (57.1%) [24], whereas in the present study, it was found in only 5.13% of the strains. Trimethoprim resistance is mediated by the expression of the enzyme DHFR (dihydrofolate reductase) and is encoded by the *dfrA1* gene that was detected in 7.69% of the strains. In general, the strains that were resistant to trimethoprim-sulfamethoxazole showed the *sul* (*sul1*, *sul2* or *sul3*) and *dfrA* (*dfrA1*, *dfrA12*) resistance genes, mainly in *S*. Derby. However, all strains were resistant to this antibiotic. This resistance may be mediated by other resistance genes, which are not assessed in this study. In *S*. Derby, *S*. *indiana*, *S*. Newport, and in two *S*. Enteritidis, the *floR* gene was detected. This gene encodes an efflux pump that confers resistance to amphenicols, which has been reported in the genomic island of *Salmonella* (SGI1) [32].

Our data are very alarming, since all of our strains came from food samples, mainly poultry meat for human consumption. Thermal processing of these products may reduce the risk of foodborne disease, but ARGs can be transferred to the gut microbiota and transfer resistance to other bacteria [33]. Therefore, our data are in line with recommendations, which confirm how important it is in the monitoring and control of antibiotic resistance to assess the presence or absence of ARGs in foodborne strains, especially in a One Health approach that recognises the circularity of human, animal, and environmental health.

## 5. Conclusions

The *Salmonella* spp. strains exhibited resistance to multiple antibiotics, as well as multiple genes associated with them. A high resistance rate to multiple antibiotics combined with multiple ARGs in isolates from raw meat, as revealed in this study, suggests that the situation is alarming in where irrational use of antibiotics is combined with inadequate surveillance and facilities to detect MDR. Continued monitoring of antimicrobial resistance in *Salmonella* strain collection along the food chain is required so that comparisons of antimicrobial resistance from the different origins can be effectively performed.

## Figures and Tables

**Table 1 antibiotics-11-00876-t001:** The *Salmonella enterica subsp. enterica* variously identified serovars isolated from meat samples of pork and poultry.

Sample of Meat	*Salmonella enterica* spp. *enterica*	Antigenic Formula	Number of Isolated Strains
pork	Enteritidis	1,9,12:g,m (without phase II)	1
poultry			22
poultry	Derby	1,4,12:f,g:-(without phase II)	5
poultry	Newport	6,8,20:e,h:1,2	5
poultry	Infantis	6,7:r:1,5	2
poultry	Kentucky	8,20:i:z_6_	2
poultry	Indiana	4,12:z:1,7	1
poultry	Mbandaka	6,7:z_10_:e,n,z_15_	1
Total	*Salmonella* spp.		n = 39

Annotation: Antigenic formula according to White-Kauffmann-Le Minor scheme somatic; somatic antigen O (1,9,12 group O9, 1,4,12; 4,12 group O4, 6,8,20; 8,20 group O8, 6,7 group O8, flagellar antigen H phase I and II.

**Table 2 antibiotics-11-00876-t002:** Multiple Antibiotic Resistance Index and phenotype pattern of *Salmonella enterica* spp. *enterica* all identified serovars isolates from meat samples of pork and poultry.

Salmonella Strains	Sample Source	Antibiotics Resistance Profiles	MAR Index
Salmonella Derby (BO4)	10 poultry	AMP-CFX-CFT-CPH-CFP-CFTI-CFQ-GEN-STR-ENR-UB-MRB-FLR-LIN/SP	0.42
	22 poultry	AMX-CPH-GEN-STR-LIN/SP-TR/SMX	0.18
	36 poultry	AMP-CFX-CFT-CPH-CFP-CFTI-CFQ-GEN-STR-ENR-UB-MRB-FLR-LIN/SP	0.42
	45 poultry	AMP-CFX-CFT-CPH-CFP-CFTI-CFQ-GEN-STR-ENR-UB-MRB-FLR-LIN/SP	0.42
	46 poultry	AMP-AMX-CFX-CFT-CPH-CFP-CFTI-CFQ-GEN-STR-ENR-UB-MRB-FLR-LIN/SP-TR/SMX	0.48
	47 poultry	AMP-AMX-CFX-CFT-CPH-CFP-CFTI-CFQ-GEN-STR-ENR-UB-MRB-FLR	0.42
Salmonella *indiana* (BO4)	61 poultry	AMX-AMX/CL-CTX-CPH-CFTI-GEN-NEO-STR-DOX-OXY-TET-FLR-LIN/SP-TR/SMX	0.42
Salmonella Infantis (CO7)	3 poultry	AMX-CPH-GEN-STR-UB-DOX-OXY-TET-FLR-LIN/SP	0.30
	38 poultry	CPH-CFTI-GEN-STR-UB-DOX-OXY-TET-FLR-LIN/SP	0.30
Salmonella Mbandaka (CO7)	9 poultry	AMP-AMX-CFX-CFT-CPH-GEN-STR-UB-LIN/SP	0.27
Salmonella Kentucky (CO8)	24 poultry	AMP-AMX-CFX-CFT-CPH-CFP-GEN-STR-ENR-UB-MRB-DOX-OXY-TET	0.42
	27 poultry	AMP-AMX-CFX-CFT-CPH-CFP-GEN-STR-ENR-UB-MRB-DOX-OXY-TET	0.42
Salmonella Newport (CO8)	1 poultry	AMP-AMX-AMX/CL-CFX-CFT-CPH-GEN-NEO-STR-ENR-UB-MRB-NOR-DOX-OXY-TET-LIN/SP	0.51
	6 poultry	AMP-AMX-AMX/CL-CFX-CFT-CPH-GEN-STR-ENR-UB-MRB-NOR-DOX-OXY-TET-LIN/SP	0.48
	8 poultry	AMP-CFX-CFT-CPH-GEN-STR-UB-MRB-DOX-OXY-TET-FLR	0.36
	12 poultry	AMP-CFX-CFT-CPH-GEN-STR-ENR-UB-MRB-DOX-OXY-TET-FLR	0.39
	13 poultry	AMP-AMX-CFX-CFT-CPH-GEN-STR-ENR-UB-MRB-DOX-OXY-TET-FLR	0.42
Salmonella Enteritidis (DO9)	2 pork	CPH-GEN-STR	0.09
	4 poultry	AMX-CPH-GEN-STR-UB-FLR-LIN/SP	0.21
	5 poultry	AMX-CPH-GEN-STR-UB-NOR-LIN/SP	0.21
	7 poultry	GEN-STR-UB-LIN/SP	0.12
	11 poultry	AMP-CFX-CFT-CPH-CFTI-GEN-STR-UB-MRB-FLR-LIN/SP	0.33
	30 poultry	CPH-GEN-STR-UB	0.12
	31 poultry	CPH-GEN-STR-UB	0.12
	32 poultry	CPH-GEN-STR-LIN/SP	0.12
	33 poultry	CPH-GEN-STR-LIN/SP	0.12
	34 poultry	CPH-GEN-STR-UB-NOR-LIN/SP	0.18
	35 poultry	CPH-GEN-STR-UB	0.12
	37 poultry	CPH-GEN-STR-UB-LIN/SP	0.15
	39 poultry	CPH-GEN-STR-UB	0.12
	40 poultry	CPH-GEN-STR-UB-LIN/SP	0.15
	41 poultry	CPH-GEN-STR-LIN/SP	0.12
	42 poultry	CPH-GEN-STR-UB-LIN/SP	0.15
	43 poultry	AMX-CPH-GEN-STR-UB-LIN/SP	0.18
	44 poultry	CPH-GEN-STR	0.09
	48 poultry	AMX-CFX-CFT-CPH-CFTI-GEN-STR-UB-FLR	0.27
	49 poultry	CPH-GEN-STR-UB	0.12
	64 poultry	CFX-CPH-GEN-NEO-STR-UB	0.18
	68 poultry	CFX-CPH-GEN-NEO-STR-UB	0.18

Letter abbreviations correspond to the individual antibiotics according to list: ampicilln (AMP), amoxicillin (AMX), amoxicillin and clavulanic acid (AMX/CL), cephalexin (CFX), cefalotin (CFT), cefapirin (CPH), cefoperazone (CFP), ceftiofur (CFTI), cefquinome (CFQ), imipenem (IPM), gentamicin (GEN), neomycin (NEO), streptomycin (STR), enrofloxacin (ENR), flumequine (UB), marbofloxacin (MRB), norfloxacin (NOR), docycycline (DOX), oxytetracycline (OXY), tetracycline (TET), florfenicol (FLR), lincomycin/spectinomycin (LIN/SP), trimethoprim-sulfamethoxazole (TR/SMX).

**Table 3 antibiotics-11-00876-t003:** Distribution of resistance genes in relation to antimicrobial resistance patterns.

*Salmonella* Strains	Sample	Phenotypic Antimicrobial Resistance Profile	Genotypic Antimicrobial Resistance Profile
*Salmonella* Derby (BO4)	10	AMP-CFX-CFT-CPH-CFP-CFTI-CFQ-GEN-STR-ENR-UB-MRB-FLR-LIN/SP	*bla*_CMY-2_, *bla*_PSE-1_, *bla*_TEM_, *aadA*, *strA*/*strB*, *floR*
	22	AMX-CPH-GEN-STR-LIN/SP-TR/SMX	*dfrA1*, *sul1*, *sul2*, *aadA*, *strA*/*strB*, *aadB*
	36	AMP-CFX-CFT-CPH-CFP-CFTI-CFQ-GEN-STR-ENR-UB-MRB-FLR-LIN/SP	*bla*_CMY-2_, *bla*_PSE-1_, *bla*_SHV_, *bla*_TEM_, *aadA*, *strA*/*strB*, *aadB*, *floR*
	45	AMP-CFX-CFT-CPH-CFP-CFTI-CFQ-GEN-STR-ENR-UB-MRB-FLR-LIN/SP	*bla*_CMY-2_, *bla*_PSE-1_, *bla*_TEM_, *dfrA1*, *dfrA1*2, *sul2*, *sul3*, *aadA*, *strA*/*strB*, *aadB*, *floR*
	46	AMP-AMX-CFX-CFT-CPH-CFP-CFTI-CFQ-GEN-STR-ENR-UB-MRB-FLR-LIN/SP-TR/SMX	*bla*_CMY-2_, *bla*_PSE-1_, *bla*_TEM_, *dfrA1*, *dfrA1*2, *sul2*, *sul3*, *aadA*, *strA*/*strB*, *aadB*, *floR*
	47	AMP-AMX-CFX-CFT-CPH-CFP-CFTI-CFQ-GEN-STR-ENR-UB-MRB-FLR	*bla*_CMY-2_, *bla*_PSE-1_, *bla*_TEM_, *aadA*, *strA*/*strB*, *floR*
*Salmonella**indiana* (BO4)	61	AMX-AMX/CL-CTX-CPH-CFTI-GEN-NEO-STR-DOX-OXY-TET-FLR-LIN/SP-TR/SMX	*bla*_CMY-2_, *bla*_TEM_, *dfrA1*, *sul1*, *sul2*, *aadA*, *strA*/*strB*, *aadB*, *aphA1*, *aphA2*, *tetA*, *tetB*, *floR*
*Salmonella* Infantis (CO7)	3	AMX-CPH-GEN-STR-UB-DOX-OXY-TET-FLR-LIN/SP	*bla*_SHV_, *aadA*, *strA*/*strB*, *tetA*, *tetB*, *floR*
	38	CPH-CFTI-GEN-STR-UB-DOX-OXY-TET-FLR-LIN/SP	*bla*_CMY-2_, *aadA*, *strA*/*strB*, *tetA*, *tetB*, *floR*
*Salmonella* Mbandaka (CO7)	9	AMP-AMX-CFX-CFT-CPH-GEN-STR-UB-LIN/SP	*bla*_PSE-1_, *bla*_TEM_, *aadA*, *strA*/*strB*
*Salmonella* Kentucky (CO8)	24	AMP-AMX-CFX-CFT-CPH-CFP-GEN-STR-ENR-UB-MRB-DOX-OXY-TET	*bla*_CMY-2_, *bla*_PSE-1_, *bla*_TEM_, *aadA*, *strA*/*strB*, *aadB*, *tetA*, *tetB*
	27	AMP-AMX-CFX-CFT-CPH-CFP-GEN-STR-ENR-UB-MRB-DOX-OXY-TET	*bla*_CMY-2_, *bla*_PSE-1_, *bla*_TEM_, *aadA*, *strA*/*strB*, *tetA*, *tetB*
*Salmonella* Newport (CO8)	1	AMP-AMX-AMX/CL-CFX-CFT-CPH-GEN-NEO-STR-ENR-UB-MRB-NOR-DOX-OXY-TET-LIN/SP	*bla*_CMY-2_, *bla*_TEM_, *aadA*, *strA*/*strB*, *aadB*, *aphA1*, *aphA2*, *tetA*, *tetB*
	6	AMP-AMX-AMX/CL-CFX-CFT-CPH-GEN-STR-ENR-UB-MRB-NOR-DOX-OXY-TET-LIN/SP	*bla*_CMY-2_, *bla*_TEM_, *aadA*, *strA*/*strB*, *tetA*, *tetB*
	8	AMP-CFX-CFT-CPH -GEN-STR-UB-MRB-DOX-OXY-TET-FLR	*bla*_PSE-1_, *bla*_TEM_, *aadA*, *strA*/*strB*, *tetA*, *tetB*, *floR*
	12	AMP-CFX-CFT-CPH-GEN-STR-ENR-UB-MRB-DOX-OXY-TET-FLR	*bla*_PSE-1_, *bla*_TEM_, *aadA*, *strA*/*strB*, *tetA*, *tetB*, *floR*
	13	AMP-AMX-CFX-CFT-CPH-GEN-STR-ENR-UB-MRB-DOX-OXY-TET-FLR	*bla*_PSE-1_, *bla*_TEM_, *aadA*, *strA*/*strB*, *aadB*, *tetA*, *tetB*, *floR*
*Salmonella* Enteritidis (DO9)	2	CPH-GEN-STR	*aadA*, *strA*/*strB*
	4	AMX-CPH-GEN-STR-UB-FLR-LIN/SP	*bla*_CMY-2_, *aadA*, *strA*/*strB*, *floR*
	5	AMX-CPH-GEN-STR-UB-NOR-LIN/SP	*bla*_SHV_, *aadA*, *strA*/*strB*
	7	GEN-STR-UB-LIN/SP	*aadA*, *strA*/*strB*
	11	AMP-CFX-CFT-CPH-CFTI-GEN-STR-UB-MRB-FLR-LIN/SP	*bla*_CMY-2_, *bla*_PSE-1_, *bla*_TEM_, *aadA*, *strA*/*strB*, *floR*
	30	CPH-GEN-STR-UB	*bla*_SHV_, *aadA*, *strA*/*strB*
	31	CPH-GEN-STR-LIN/SP	*bla*_SHV_, *aadA*, *strA*/*strB*
	32	CPH-GEN-STR-LIN/SP	*bla*_SHV_, *aadA*, *strA*/*strB*
	33	CPH-GEN-STR-UB-NOR-LIN/SP	*bla*_SHV_, *aadA*, *strA*/*strB*
	34	CPH-GEN-STR-UB	*bla*_CMY-2_, *aadA*, *strA*/*strB*
	35	CPH-GEN-STR-UB-LIN/SP	*bla*_SHV_, *aadA*, *strA*/*strB*
	37	CPH-GEN-STR-UB	*bla*_SHV_, *aadA*, *strA*/*strB*
	39	CPH-GEN-STR-UB	*bla*_SHV_, *aadA*, *strA*/*strB*
	40	CPH-GEN-STR-UB-LIN/SP	*bla*_TEM_, *aadA*, *strA*/*strB*
	41	CPH-GEN-STR-LIN/SP	*bla*_TEM_, *aadA*, *strA*/*strB*
	42	CPH-GEN-STR-UB-LIN/SP	*bla*_SHV_, *aadA*, *strA*/*strB*
	43	AMX-CPH-GEN-STR-UB-LIN/SP	*bla*_SHV_, *aadA*, *strA*/*strB*
	44	CPH-GEN-STR	*bla*_CMY-2_, *aadA*, *strA*/*strB*
	48	AMX-CFX-CFT-CPH-CFTI-GEN-STR-UB-FLR	*bla*_CMY-2_, *aadA*, *strA*/*strB*, *floR*
	49	CPH-GEN-STR-UB	*bla*_SHV_, *aadA*, *strA*/*strB*
	64	CFX-CPH-GEN-NEO-STR-UB	*bla*_TEM_, *aadA*, *strA*/*strB*, *aphA1*, *aphA2*
	68	CFX-CPH-GEN-NEO-STR-UB	*bla*_TEM_, *aadA*, *strA*/*strB*, *aphA1*, *aphA2*

Letter abbreviations correspond to the individual antibiotics according to list: ampicilln (AMP), amoxicillin (AMX), amoxicillin and clavulanic acid (AMX/CL), cephalexin (CFX), cefalotin (CFT), cefapirin (CPH), cefoperazone (CFP), ceftiofur (CFTI), cefquinome (CFQ), imipenem (IPM), gentamicin (GEN), neomycin (NEO), streptomycin (STR), enrofloxacin (ENR), flumequine (UB), marbofloxacin (MRB), norfloxacin (NOR), docycycline (DOX), oxytetracycline (OXY), tetracycline (TET), florfenicol (FLR), lincomycin/spectinomycin (LIN/SP), trimethoprim-sulfamethoxazole (TR/SMX).

**Table 4 antibiotics-11-00876-t004:** Description of primer sets, annealing temperature and product size for the molecular gene identification [20,21,22].

Multiplex PCR or Single PCR	Gene/Antibiotic	Primer Sequences 5’-3’	Annealing Temperature	Product Size (bp)
Multiplex 1	*aadA* streptomycin	F-GTG GAT GGC GGC CTG AAG CC R-AAT GCC CAG TCG GCA GCG	63 °C	525 bp
Multiplex 1	*strA/strB* streptomycin	F-ATG GTG GAC CCT AAA ACT CT R-CGT CTA GGA TCG AGA CAA AG	63 °C	893 bp
Multiplex 2	*aphA1* neomycin	F-ATG GGC TCG CGA TAA TGT C R-CTC ACC GAG GCA GTT CCA T	55 °C	634 bp
Multiplex 2	*aphA2* neomycin	F-GAT TGA ACA AGA TGG ATT GC R-CCA TGA TGG ATA CTT TCT CG	55 °C	347 bp
Multiplex 2	*aadB* gentamicin	F-GAG GAG TTG GAC TATGGA TT R-CTT CAT CGG CAT AGT AAA AG	55 °C	208 bp
Multiplex 3	*tetA* tetracycline	F-GGC GGT CTT CTT CAT CAT GC R-CGG CAG GCA GAG CAA GTA GA	63 °C	502 bp
Multiplex 3	*tetB* tetracycline	F-CGC CCA GTG CTG TTG TTG TC R-CGC GTT GAG AAG CTG AGG TG	63 °C	173 bp
Multiplex 4	*sul1* sulfamethoxazole	F-CGG CGT GGG CTA CCT GAA CG R-GCC GAT CGC GTG AAG TTC CG	66 °C	433 bp
Multiplex 4	*sul2* sulfamethoxazole	F-CGG CAT CGT CAA CAT AAC CT R-TGT GCG GAT GAA GTC AGC TC	66 °C	721 bp
Single PCR	sul3 sulfamethoxazole	F-GGGAGCCGCTTCCAGTAAT R-TCCGTGACACTGCAATCATTA	60 °C	500 bp
Single PCR	*dfrA1* trimethoprim	F-CAATGGCTGTTGGTTGGAC R-CCGGCTCGATGTCTATTGT	62 °C	253 bp
Single PCR	*dfrA10* trimethoprim	F-TCAAGGCAAATTACCTTGGC R-ATCTATTGGATCACCTACCC	59 °C	433 bp
Single PCR	*dfrA12* trimethoprim	F-TTCGCAGACTCACTGAGGG R-CGGTTGAGACAAGCTCGAAT	63 °C	330 bp
Single PCR	*floR* florfenicol	F-CACGTTGAGCCTCTATATGG R-ATGCAGAAGTAGAACGCGAC	61 °C	888 bp
5	*bla*_TEM_ ampicillin	F-TTAACTGGCGAACTACTTAC R-GTCTATTTCGTTCATCCATA	55 °C	247 bp
5	*bla*_SHV_ ceftiofur	F-AGGATTGACTGCCTTTTTG R-ATTTGCTGATTTCGCTCG	55 °C	393 bp
5	*bla*_CMY-2_ ceftiofur	F-GACAGCCTCTTTCTCCACA R-TGGACACGAAGGCTACGTA	55 °C	1000 bp
Single PCR	*bla*_PSE-1_ ampicillin	F-GCAAGTAGGGCAGGCAATCA R-GAGCTAGATAGATGCTCACAA	60 °C	461 bp
Single PCR	*bla* _CTX-M_	F-CGCTTTGCGATGTGCAG R-ACCGCGATATCGTTGGT	60 °C	585 bp

## Data Availability

The data presented in this study are available on request from the corresponding author. The data are not publicly available due to their containing information that could compromise the image of the meat processing plants.

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
