# Peer review of "Multi-Drug Resistance to Salmonella spp. When Isolated from Raw Meat Products"

_antibiotics, 2022, doi:10.3390/antibiotics11070876_

Round 1

Reviewer 1 Report

This manuscript demonstrated the Salmonella isolates’ MDR profile that originated from poultry and pork. The design of the study is clear. Only the presenting of the results needs some improvements. For all the tables, horizontal dividing lines are not showed, which makes them difficult to read. Although this may not be a big deal. But in Table 3, the Salmonella Derby part is very chaos and the difference in alignment can be obviously seen from the Salmonella Kentucky part.

Still about data presenting, on page 3 line 99 to 119, these results can have a better layout in a table.

One minor careless mistake in writing is on page 4 line 147, the “Table …” needs to be clarified.

Author Response

Thank you for all your suggestion.

 For all the tables, horizontal dividing lines are not showed, which makes them difficult to read. Although this may not be a big deal. But in Table 3, the Salmonella Derby part is very chaos and the difference in alignment can be obviously seen from the Salmonella Kentucky part.

Thank you for your suggestion. We were correct the table, especially 3 and we hope that it is more clear.

One minor careless mistake in writing is on page 4 line 147, the “Table …” needs to be clarified.

Thank you for your suggestion. We were remove mistake on page 4  line 147.

Reviewer 2 Report

The aim of this research is to determine the antibiotic resistance of Salmonella spp. isolated from raw meat products from beef, pork, and poultry production plants. The study is interesting and the manuscript is well written. Here are some comments for more improvement to the manuscript:

- Name of the pathogens needs to be in italic.

- Line 56: (Antibiotics 2022 in press). please provide a reference for a published paper.

- Provide the class of each antibiotic and group then according to their class.

-  Line 61: add a reference.

- In section 4.3: provide the concentration of each antibiotic.

- please provide a correlation between phenotypic and genotypic analysis as shown in this paper https://www.mdpi.com/2079-6382/10/12/1450

- Why do you report the presence of Citrobacter braakii in your samples

-  Explain the Antigenic formula in the table footnote.

- Line 136: provide the name of the classes.

- In tables footnote: add the full name of the antibiotics.

Author Response

Thank you for all your suggestion.

 Name of the pathogens needs to be in italic.

We were correct this mistake.

- Line 56: (Antibiotics 2022 in press). please provide a reference for a published paper.

We added correct references.

Joanna PÅ‚awiÅ„ska-Czarnak, Karolina Wódz, Lidia Piechowicz, Ewa Tokarska-Pietrzak, Zbigniew BeÅ‚kot, Janusz Bogdan, Jan WiÅ›niewski, Piotr KwieciÅ„ski, Adam KwieciÅ„ski, Krzysztof Anusz. Wild Duck (Anas platyrhynchos) as a Source of Antibiotic-Resistant Salmonella enterica subsp. diarizonae O58—The First Report in Poland. Antibiotics 2022, 11, 530. https://doi.org/10.3390/antibiotics11040530

- Provide the class of each antibiotic and group then according to their class.

We were provide the class of antibiotic.

-  Line 61: add a reference.

We were add reference.

- In section 4.3: provide the concentration of each antibiotic.

We were add concentration of each antibiotic.

- please provide a correlation between phenotypic and genotypic analysis as shown in this paper https://www.mdpi.com/2079-6382/10/12/1450

We are sorry but we are unable to show correlation between phenotypic and genotypic antibiotic resistance as Authors https://www.mdpi.com/2079-6382/10/12/1450 on image 3 because we do not have software for such visualization

- Why do you report the presence of Citrobacter braakii in your samples

In first part of this study we identified amongst samples, colonies looks like Salmonella. After biochemical and molecular diagnostic we identified these strain as Citrobacter braakii  - PÅ‚awiÅ„ska-Czarnak, J.; Wódz, K.; Kizerwetter-Åšwida, M.; Nowak, T.; Bogdan, J.; KwieciÅ„ski, P.; KwieciÅ„ski, A.; Anusz, K. Citrobacter braakii Yield False-Positive Identification as Salmonella, a Note of Caution. Foods 2021, 10, 2177. doi: 10.3390/foods10092177

-  Explain the Antigenic formula in the table footnote.

We were add annotation below table

- Line 136: provide the name of the classes

We were add name of the classes of antibiotics.

- In tables footnote: add the full name of the antibiotics.

We were add full name of the of antibiotics.

Round 2

Reviewer 2 Report

The authors addressed all the concerns. However, there are still few grammatical/typo errors that will need to be corrected